# Gut Microbiota and New Microbiome-Targeted Drugs for *Clostridioides difficile* Infections

**DOI:** 10.3390/antibiotics13100995

**Published:** 2024-10-20

**Authors:** Ahran Lee, Jung Sik Yoo, Eun-Jeong Yoon

**Affiliations:** Division of Antimicrobial Resistance Research, Korea National Institute of Health, Korea Disease Control and Prevention Agency, Cheongju-si 28159, Republic of Korea

**Keywords:** *Clostridioides diffcile*, *Clostridioides diffcile* infections, gut microbiota dysbiosis, microbiota-based therapeutics

## Abstract

*Clostridioides difficile* is a major causative pathogen for antibiotic-associated diarrhea and *C. difficile* infections (CDIs) may lead to life-threatening diseases in clinical settings. Most of the risk factors for the incidence of CDIs, i.e., antibiotic use, treatment by proton pump inhibitors, old age, and hospitalization, are associated with dysbiosis of gut microbiota and associated metabolites and, consequently, treatment options for CDIs include normalizing the composition of the intestinal microbiome. In this review, with an introduction to the CDI and its global epidemiology, CDI-associated traits of the gut microbiome and its metabolites were reviewed, and microbiome-targeting treatment options were introduced, which was approved recently as a new drug by the United States Food and Drug Administration (U.S. FDA), rather than a medical practice.

## 1. Introduction

*Clostridioides difficile* is a Gram-positive, spore-forming, and obligate anaerobic bacteria [1]. *C. difficile* was first reported as a part of normal flora composing gut microbiota in healthy newborns in the 1930s [1] and it was recognized as a cause of antibiotic-associated diarrhea to colitis in the late 1970s through a screening of all-age patients [2]. The *C. difficile* infection (CDI) is a well-known healthcare-associated infection and it also causes community-associated infections [3].

*C. difficile* is part of the normal gut microbiota, though the bacterial species occupy only a limited portion of the gut microbiome. Pathogenic *C. difficile* entry into the gastrointestinal (GI) tract and its ability to form spores resistant to harsh conditions allow the bacterial pathogen to be persistent in the gut. Germination of the spores starts in the imbalanced gut microbiome for varied reasons, and proliferation of the pathogen is started [4]. Known risk factors are associated with developing CDI and the risk factors are somehow linked to gut microbiota dysbiosis [5], including exposure to antimicrobial agents, old age, medication with proton pump inhibitors, and long-term hospitalization [6]. Approximately 2% of community people and from 20% to 30% of hospitalized adults are asymptomatic carriers [7]. Gut microbiota dysbiosis leads to immunological and biochemical disruptions, resulting in increased colonization capacity and favoring the germination and proliferation of *C. difficile* [8]. Due to the exotoxins, which are produced by toxigenic clones of the pathogen, severe CDIs can lead to pseudomembranous colitis and toxic megacolon, which could be lethal in extreme cases [9], and the incubation period varies from a couple of days to several weeks [7,10].

CDI is associated with high morbidity and mortality, leading to a high socioeconomic burden on the healthcare system. As well, its high rate of recurrence elevates the cost compared to the initial incidence of the infectious disease [11,12]. In addition, during the COVID-19 pandemic, mortality rates of CDI patients were increased and subsequent per-patient cost was also elevated [13]. Research and development of therapeutic options for the disease is focused on repeated recurrent CDI, mostly by restoring microbiota [14,15,16].

This review aims to present CDI-associated traits of the gut microbiome and treatment options and to highlight the microbiota-based medications, which were approved recently as a new drug by the United States Food and Drug Administration (U.S. FDA), will be introduced with a detailed comparison.

## 2. Epidemiology and Pathophysiology of CDI

Among the pathogens causing all types of healthcare-associated infections in the USA, *C. difficile* is a leading pathogen (12.1%), followed by *Staphylococcus aureus*, *Klebsiella*, and *Escherichia coli* [17]. And a high rate of CDI incidence is a significant threat to public health and an economic burden for healthcare [3,18]. The mortality associated with CDI was also significant, though it was less than 2% of total patients with CDI until 2000, before the global outbreak of the toxic clone ribotype (RT) 027. And the mortality rate was recorded as up to 17% for outbreak cases and 5% for non-outbreak periods in general [19,20,21,22,23].

The incidence of CDI continuously increased until the early 2000s, though it varied by country, and the augmentation was mostly stopped, drawing a plateau graph after the 2010s [3,24,25,26]. According to the reports of the current surveillance study, the incidence of CDI in the U.S. has been estimated as 8.3 cases per 10,000 patient days by a meta-analysis using the data between 2010 and 2019 [27], and the increase in morbidity, recurrence, prolonged hospitalization, and mortality resulted in the substantial economic burden of CDIs, which is estimated to be USD 5.4 billion annually [28]. In Europe, the incidence of healthcare-associated CDIs was 2.58 cases per 10,000 in 2020 [29], 5.4 cases per 10,000 patient days between 2000 and 2016 in Asian countries [30], and, finally, in South Korea, 18 cases per 10,000 patient days were estimated in 2020 [31].

Especially among healthcare-associated CDI, recurrent CDI, which indicates the CDI after an initial episode of CDI usually within 30 days after the initial diagnosis of CDI, is one of the sources of morbidity for patients. Globally, 20 to 35% of CDI patients experience at least one recurrent episode [32].

The global spread of CDI in healthcare facilities was highly associated with a hypervirulent RT027 clone, which primarily emerged in Canada, leading to about 2000 deaths [33,34], and subsequent dissemination throughout North America and, further, to Europe in 2005 [35,36]. In the USA, the RT027 clone was dominant until the early 2010s, taking 35% of the *C. difficile* clinical isolates in 2011, followed by RT014/020 and RT106 [37]. However, in 2016, the RT106 became the primary prevalent clone, taking 15% of all the screened clinical isolates, and the RT027 and R014/020 clones followed [37]. In European countries, the RT014/020 clone was dominant, taking 17.7% of the clinical isolates recovered in 2018 and 2019, and R002 and R079 clones were followed, though the proportions differed by country. The geographic and genetic distances among the prevalent clones suggest varied ways to disseminate the pathogen [38].

In addition to the healthcare-associated infections, community-associated CDI cases are increasingly reported globally [3,18,39,40], together with its association with the clones of animal and environmental origin in One Health aspects, such as RT014/020, RT002, and R001, and so on [41], while the transmission via the food chain lacks evidence [42]. Traits in the composition of *C. difficile* clones are alike between both the healthcare-associated and the community-associated infections [40] and the community-associated infections present milder symptoms than the cases of healthcare-associated infections [43].

Since *C. difficile* is an obligate anaerobe, the bacterial cell is not able to survive in the aerobic condition. Consequently, its persistence and transmission rely on the formation of endospores, which are aerotolerant and highly resistant to harsh environments [44]. *C. difficile* sporulation is triggered by environmental stimuli including quorum sensing [45] and nutrient limitations [46]. And the germination of *C. difficile* spores is activated by defined environmental signals that trigger the return of metabolic activity and the outgrowth of the vegetative cell [46,47].

In addition, *C. difficile* produces two clostridial toxins, i.e., TcdA, an enterotoxin associated with inflammation and tissue necrosis at the intestinal tract, and TcdB, a cytotoxin, leading to the formation of pseudomembrane [48,49], and a binary toxin *C. difficile* transferase (CDT). All three toxins are produced under tight regulation; however, mutations in the regulatory system of the toxin may cause excessive production of the toxin resulting in a bad prognosis for the patients. The epidemic *C. difficile* clone RT027, the NAP1/BI/027, was a good example of a hypertoxic clone by overproduction of TcdB by a mutation in the anti-sigma factor TcdC, resulting in a distinguished hypervirulent phenotype of the clone [50].

## 3. Gut Environment Including Microbiota and Its Association with CDI Incidence

As mentioned above, vegetative cells of *C. difficile* are susceptible to oxygen and varied antimicrobials, but spores of the bacterial species can survive for long periods in harsh conditions, such as gastric intestinal tract with extreme pH fluctuation, and can easily reach the intestine, the target site [51]. Germination of *C. difficile* spores and the growth of vegetative cells occur in the lower gastrointestinal tract having a lower concentration of oxygen [52]. Spores of *C. difficile* are germinated into vegetative cells by primary bile acid [43]. Healthy gut microbiota provides colonization resistance against *C. difficile* by varied mechanisms including the production of inhibitory metabolites, such as secondary bile acids [53], short-chain fatty acids (SCFAs) [54], and antimicrobials [55], as well as competition for nutrients, especially proline and other amino acids necessary for Stickland fermentation [56].

### 3.1. Gut Microbiota

The gastrointestinal tract contains a rich microbial community composed of 100 trillion microorganisms, compared with other body parts of human beings [57]. The gut microbiota is a dynamic and complex community of microorganisms [58,59]. The composition of the gut microbiota is mostly determined by diet [60], and it has large interpersonal differences regardless of the genetic closeness between human hosts [61]. The gut microbiota in healthy human beings is stable, resilient, and symbiotic interaction with the host and the microbial community presents high taxonomic diversity, high microbial gene richness, and stable core microbiota [62]. And criteria for a healthy gut microbiome is difficult, except for the high diversity of the composition [63].

Although no more than a high diversity has been generally known to be associated with a healthy gut microbiota, they contribute functionally to the host’s health by helping with (i) protection against pathogens, (ii) epithelial barrier integrity, (iii) digestion and nutrition absorption, and the (iv) development of immune system [61]. Major bacterial phylum in the human gut are Bacteroidetes, Firmicutes, Actinobacteria, and Proteobacteria, as the primary two presented the highest abundance [58]. The composition of the human gut microbiome changes relative to age: Bacteroides followed by Bifidobacteria in infancy, Firmicutes followed by Bacteroides in adolescence, Bacteroides followed by Firmicutes in adulthood, and obligate and facultative anaerobes in old age [64].

*C. difficile* is considered a member of the normal gut microflora, but not dominant compared to the other anaerobes. Rates of colonization in the human gut for *C. difficile* are different for age groups—highest in early infancy and decreases with age [65,66]—and such gut colonization is not always associated with the incidence of infections. Incidence of CDI is rather associated with disruption of the gut microbiota. Varied risk factors are linked with the infection cases. For the initial CDIs, antimicrobial exposure, old age of over 65 years, long-term hospitalization, acid-suppression medications, immunosuppression, abdominal surgery or nasogastric tube, diabetes mellitus, end-stage renal disease, and inflammatory bowel diseases are associated; and, for the recurrent CDIs, together with all the risk factors for initial CDIs, prior infection with a hypervirulent strain is associated [67].

Disruption in gut microbiota weakens the intestinal defense owing to the healthy microbiota and makes the gastrointestinal tract susceptible to foreign pathogens including *C. difficile* [68,69,70]. Such dysbiosis is mainly caused by the usage of antimicrobials, mostly those having a broad spectrum of activity, resulting in monotoned gut microbiota with antimicrobial resistance. The imbalanced composition of the gut microbiota, together with its association with CDI, is differed by the antimicrobial drugs [71,72], and varied drug classes have been indicated, such as clindamycin, fluoroquinolones, and third- and fourth-generation cephalosporins [73,74]. Since *C. difficile* spores are resistant to antimicrobial drugs, the bacterial species gains a competitive advantage with the loss of competitive inhibitors and nutrition competitors, allowing a new ecologic niche for better proliferation of *C. difficile* [75].

Patients with primary CDI and those with recurrent CDI had different dynamics of gut microbiome [68,76]. As decreased diversity of the microbiota is the common traits of the gut microbiota of CDI patients, a portion of the phylum Bacteroidetes were decreased with the families *Bacteroidaceae* and Clostridiales Incertae Sedis XI, while the portion of *Enterococcaceae* belonging to Firmicutes was enriched in CDI patients [77]. Similarly, reduction in the portion of Bacteroides and augmentation of the proportion of Firmicutes including *Enterococcaceae,* and Proteobacteria including Enterobacteriaceae in the gut microbiota of CDI patients [78]. In the CDI animal model, decreased Firmicutes, together with an augmenting abundance of Proteobacteria, especially *Enterobacteriaceae,* were observed [79]. Gut dysbiosis may increase CDI susceptibility by shifting many aspects of colonization resistance, including competition for nutrients, antimicrobial production, and bile acid metabolism [80].

### 3.2. Gut Microbiota Metabolites

Various metabolites produced by a human host and gut microbiota may interact with *C. difficile*. Primarily, SCFAs, which are produced by microbial fermentation, were reported to control the pH in the intestinal lumen, and it is known to be associated with *C. difficile* colonization [80]. In addition, butyrate, which is produced while the SCFAs are produced, is known to play an important role in the regulation of growth, intestinal epithelial cell diversity, and preserving the integrity of the gut epithelium [81,82]. SCFAs act the stimulation of the mucin and antimicrobial peptides production and the defense barriers and it can protect the host from infection [83]. Notably, the incidence of CDI is associated with a decrease in butyrate-producing bacteria and an increase in lactic acid-producing bacteria [84]. As well, SCFAs inhibit *C. difficile* development and have an anti-inflammatory effect [85]. Consequently, butyrate-producing bacteria including *Lachnospiraceae* and *Ruminococcaceae* families were reported to be considerably decreased in stool specimens for patients with diarrhea and CDI [85,86].

In humans, primary bile acids are synthesized from cholesterol in the liver, secreted into the bile, and, after they are expelled in the intestine, bio-transformed in the colon to the secondary bile acids by the gut microbiota consortium [87,88]. When antibiotic exposure causes gut dysbiosis, the primary bile acids are not transformed into secondary bile, and the excess primary bile acids, i.e., cholic acid derivatives, lively germinate the *C. difficile* spores in the small intestine [89], while secondary bile acids, i.e., deoxycholic acid, in low concentrations of large intestine inhibit vegetative growth of *C. difficile* [51]. The primary bile acid, chenodeoxycholate and secondary bile acids, deoxycholic acid and lithocholic acid, inhibit the germination of spores and the growth of *C. difficile* in healthy microbiota [90].

Some amino acids also play a significant role in the germination of *C. difficile*. Glycine, in combination with certain bile acids, initiates the germination of *C. difficile* [89]. In addition, histidine, arginine, aspartic acid, and valine can further activate germination in the existence of both glycine and conjugated bile acids [91,92].

The availability of carbohydrates in the gut can be affected by antibiotics in several ways since a decrease in the diversity of the gut microbiota community reduces the competition for limited nutrients [93]. Succinate is an organic acid produced through microbial carbohydrate fermentation of dietary carbohydrates by specific bacterial species including Bacteroides and Negativicutes [94], and *C. difficile* is one of the bacterial species thriving on the presence of succinate [95], while higher concentrations of succinate are toxic to the gut cells and gut microbiota [96]. Sialic acid, a host mucosal monosaccharide, is also a nutrient source of *C. difficile* and antimicrobial dysbiosis, leading to the over-liberation of sialic acids contributes to the outgrowth of *C. difficile*, which do not produce sialidases but can metabolize free sialic acid [97].

## 4. Medications for CDI

CDIs were suspected in any patient with diarrhea, who has been treated by using antibiotics within the previous 3 months, has been recently hospitalized, and/or has an occurrence of diarrhea within 48 h or more after hospitalization [98]. Not only hospitalized patients, but community people without previous hospitalization or antibiotic exposure can have diarrhea through CDI [99]. To diagnose CDI, molecular testing using PCR methods, antigen testing mostly for the presence of *C. difficile* antigen glutamate dehydrogenase, toxin testing using tissue culture cytotoxicity assay for toxin B or enzyme immunoassay for toxin A and toxin B, and stool cultures, mostly the toxigenic cultures, are used [100,101].

The initial CDI episodes, antimicrobial medications are recommended with strong evidence by the clinical practice guidelines from the Infectious Disease Society of America (IDSA), the Society for Healthcare Epidemiology of America (SHEA), and the European Society of Clinical Microbiology and Infectious Diseases (ESCMID). In addition, regardless of the severity, fidaxomicin or vancomycin were suggested as first-line treatment: a 10-day b.i.d. oral administration of fidaxomicin 200 mg for both non-severe and severe episodes of CDI and, alternatively, 10-day q.i.d. oral administration of vancomycin 125 mg, and, if both are unavailable, 10-day t.i.d. oral administration of metronidazole 500 mg (Table 1) [100,102].

For recurrent CDI, usage of antimicrobials is still recommended until the second recurrence. Details of medication, such as antimicrobials, dosage, duration, and route of administration, depend on the usage of antimicrobial drugs used for the initial CDI. And, from the third recurrence of CDI with several failed appropriate antimicrobial treatments, fecal microbiota transplantation (FMT) was strongly recommended for the treatment with moderate quality of evidence. And, since 2021, FMT has been recommended for patients with multiple recurrent CDIs [67,100,102].

To cumulate better evidence for potential options for CDI intervention, clinical trials were conducted for varied types of candidate agents (Figure 1). According to ClinicalTrials.gov as of 4 September 2023, a total of 146 clinical trials were registered as new studies targeting CDI, including initial and recurrent cases. In 2002, just before the deadly emergence of CDIs in North America, the first clinical trial, a phase II for CDI treatment of prospective, randomized double-blind design for the activity of nitazoxanide to compare CDI to metronidazole was started in the USA, showing that the nitazoxanide was as effective as metronidazole, though it is used for diarrhea caused by parasites [103]. After the first study, clinical trials for chemical-based small molecules were continued (Figure 1). The efficacy and safety of FMT were studied in recurrent CDI by clinical trials in 2012 and peaked in 2013. And clinical trials for microbiota were started in 2010 and have continued up to now. The most studied type of intervention was FMT (n = 60), and chemical-based small molecule (n = 58) and microbiota (n = 18) followed. Notably, the FMT was mostly sponsored by others (53/60, 88.3%) outside of the industry, while the chemicals (42/58, 72.4%) and microbiota (13/18, 72.2%) were mostly sponsored by industry. As the deadly emergence of CDI occurred in North America, 61% of the clinical studies were conducted in North America, followed by Europe (18%) and Asia (8%).

## 5. Operational Treatment Using Fecal Microbiota Transplantation

Several observational studies and clinical trials demonstrated high efficacy of FMT within eight weeks with or without comparison with the standard of care using antimicrobial therapy with varied efficacy by delivery method and by number of administrations: delivery by lower gastrointestinal endoscopy was superior to all other methods for delivery and repeated FMT increased the efficacy significantly [104]. While significant adverse effects related to FMT are very rare, less than 1%, the immunocompromised patients and the patients with underlying gastrointestinal diseases could be susceptible to a life-threatening adverse effect of FMT and minor adverse events, such as diarrhea, constipation, abdominal pain, nausea, and vomiting, could be occurring in about 1% of the patients [105].

Considering the basic concept of FMT for the treatment of CDIs, rebuilding the disrupted gut microbiota in a short time to restore its function for gut homeostasis and barrier defense [106,107], FMT is an ideal therapeutic option for CDI, which has a high rate of relapse, 20% to 30% [39]. And, consequently, the FMT presented a definite advantage in recurrent CDI treatment [108]. Though FMT is known to be generally safe and well-tolerated [109], FMT still bears several risks, such as the possible transfer of pathogenic microorganisms through the operation, less reproducibility relies on the fecal donors [110], and operational difficulties, which are mostly by colonoscopy, nasogastric or nasoduodenal tube [111,112], and mostly linked with minor adverse events.

## 6. U.S. FDA-Approved Live Biotherapeutic Products for Recurrent CDI

In recent years, due to the concerns with FMT mentioned above, pharmaceutical companies tried to develop standardized FMT products to reduce variability, ensure safety, and improve performance and scalability. Thus, pharmaceutical products containing live bacteria, so-called live biotherapeutic products, were developed to cure CDI [113], and the live biotherapeutic products are recognized by the FDA as a class of biological drug products [114], not an investigational medical procedure [115]. For marketable and standardized microbiome-based products, healthy stool donors need to be controlled as a part of active ingredients and good manufacturing practices need to be settled down as required by the FDA [114,116]. At this time, two live biotherapeutic products have been approved by the U.S. FDA for the prevention of rCDI: REBYOTA^TM^ (formerly RBK2660), a fecal microbiota suspension, live-jslm, and VOWST^TM^ (formerly SER-109), fecal microbiota spores, and live-brpk (Table 2).

### 6.1. REBYOTA^TM^

REBYOTA^TM^, previously the RBX2660, is the first fecal microbiota product approved by the FDA on 30 November 2022 [117]. It is a rectally administered suspension of live fecal microbiota for prevention of recurrence of CDI in individuals over 18 years old after the completed antimicrobial treatment for recurrent CDI [118]. Since FMT is known to be a well-recognized strategy to manage recurrent CDI by restoring gut microbiota [119], safe and effective application of the human-derived microbiota-based live biotherapeutic product was developed for the disease by Ferring Pharmaceutical Inc. and the REBYOTA^TM^ is the results. The REBYOTA^TM^ needs a single-dose rectal administration to the patients and the treatment by single-dose REBYOTA^TM^ administration was demonstrated to expect better safety and efficacy than that by double dose [120].

A phase III trial of a single dose of REBYOTA^TM^ was demonstrated to have an efficacy of 70.6% in eight weeks in subjects with more than two recurrences of CDI and over 90% of subjects with treatment success had sustained response through six months compared to the placebo [121]. From a 24-month phase II clinical trial for REBYOTA^TM^, long-term safety was also demonstrated with guaranteed efficacy, and, notably, there were no reported infections caused by pathogens traceable to [120,122]. Of all the adverse events, 84% were mild to moderate and the most common symptom was diarrhea reported in 30% of subjects [120]. While 13% of subjects experienced urinary tract infections, none were associated with REBYOTA^TM^ itself or the enema procedure [120]. Two severe adverse events associated with the REBYOTA^TM^ treatment led to the death of one participant [120].

Through several clinical trials for REBYOTA^TM^, effective restoration of the gut microbiota was observed [120,123,124], and the mechanism of action of REBYOTA^TM^ was considered to be restoration of the gut microbiome against dysbiosis.

REBYOTA^TM^ is a combination product for rectal administration. The manufacturing process is implemented after the collection of a stool donation from a single donor by combining the donor stool with a cryoprotectant excipient solution. The condition of the stool donor is screened through a qualification process that includes initial screening and monitoring. A 150 mL suspension of the fecal microbiota is filled into a 250 mL ethylene vinyl acetate (EVA) bag to produce the REBYOTA^TM^ product to contain between 1 × 10^8^ and 5 × 10^10^ CFU/mL of fecal microbes, including >1 × 10^5^ CFU/mL of Bacteroides [110].

Each EVA bag is attached with a temporary label, stored at −80 °C under quarantine, and released after the receipt of acceptable donor testing results. The expiry date for the final drug product is 36 months from the date of manufacture when stored at −80 °C. The drug product is stable for 96 h when stored at refrigerated conditions after a 24 h thaw [118].
antibiotics-13-00995-t002_Table 2Table 2FDA-approved live biotherapeutic products for recurrent CDI.ProductRebyota^TM^ [125,126]Vowst^TM^ [127]TypeFecal microbiota, live-jslmFecal microbiota spores, live-brpkManufacturerFerring PharmaceuticalsSeres therapeuticsindicationRecurrent CDIRecurrent CDIDosage form150 mL of liquid suspension administered rectallyCapsule for oral administrationActive ingredients1 × 10^8^ to 5 × 10^10^ CFU/mL of fecal microbes including >1 × 10^5^ CFU/mL of Bacteroides1 × 10^6^ to 3 × 10^7^ sCFU/capsule of ca. 50 species of Firmicutes sporesExcipientsPolyethylene glycol 3350 and 0.9% sodium chlorideGlycerol in 0.9% sodium chlorideStorageStorage at −80 °CThawed in refrigerator (4 °C) prior to administration and stable at room temperature for up to 2 daysRoom temperature (2 °C to 25 °C)AdministrationRectal administration of 150 mLOral administration of 4 capsules each day for 3 consecutive daysCostUSD 9 K to 10 K for a single 150 mL doseUSD 17.5 K for 12 capsules

### 6.2. VOWST^TM^

VOWST^TM^, previously named Ser-109, is the first U.S. FDA-approved encapsulated fecal microbiota on 26 April 2023 allowing oral administration [128]. The indication for VOWSTTM is to prevent the recurrence of CDI [129] and the dose, oral administration of four capsules for three consecutive days, was determined for recurrent treatment by the phase III clinical trial [125,130] over the phase II clinical trial, showing no efficacy for the prevention of recurrent CDI by single day administration of the dose [126].

VOWST^TM^ is encapsulated 1 × 10^6^ to 3 × 10^7^ spores of ca. 50 species of Firmicutes bacteria colony forming unit. The shelf life for the final drug product is 36 months from the date of manufacture stored at 2° to 25 °C [129].

A double-blind placebo-controlled phase III clinical trial demonstrated that the CDI recurrence was significantly lower in the VOWST^TM^-treated group than in the placebo group at 12% vs. 40% with acceptable adverse events of mild to moderate symptoms, mostly the gastrointestinal disorders [125,130]. Notably, for the patients treated by using VOWST^TM^, the composition of gut microflora was changed to decline in *Enterobacteriaceae* and to increase in Firmicutes, including *Ruminococcaceae* and *Lachnospiraceae* [130], supporting the possible mechanism of action, and restoring the gut microbiota.

## 7. Conclusions

The incidence of CDIs increases and the socioeconomic burden of the diseases is growing. As antimicrobial exposure is one of the major risk factors for CDI incidence, more understanding is needed of the relationship between the gut microbiota environment and CDI incidence and the development of new therapeutic options, preferably an approved medication, needs to be developed. Since the most difficult point of the disease is the high rate of relapse, many researchers continued to find therapeutic options to be out of the loop of recurrent CDI and the REBYOTA^TM^ and VOWST^TM^ were good starts as a fundamental solution with the least recurrence. A comprehensive understanding of gut microbiota composition and dysbiosis is continually needed for the development of further approaches for microbiota manipulation and targeted therapies.

## Figures and Tables

**Figure 1 antibiotics-13-00995-f001:**
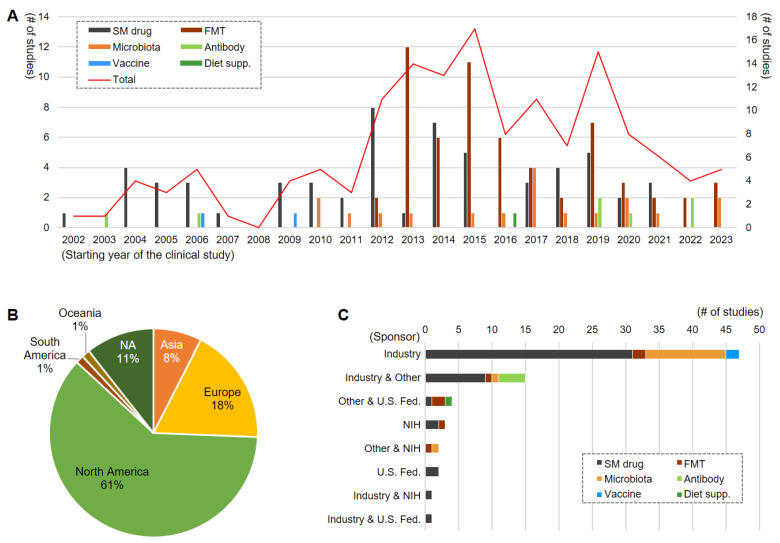
Schematic presentation of the 146 clinical trials to develop therapeutic options for CDI. Analyzed the data from http://www.ClinicalTrials.gov, accessed on 4 September 2023. (**A**) The year of starting the clinical study by category of the therapeutic interventions, (**B**) continent-based locations running the clinical study (multiple choice was made if the study was conducted in multiple continents), (**C**) sponsor types of the clinical trials by category of the therapeutic interventions. A total of 146 clinical trials targeting CDI started before 4 September 2023, extracted from ClinicalTrials.gov, were analyzed. SM drug, small molecule drug; FMT, fecal microbial transplantation; diet supp., diet supplement; U. S. Fed., The United States Federal; NIH, National Institute of Health; NA, not available to accept the data.

**Table 1 antibiotics-13-00995-t001:** Recommendations for the treatment of CDI in adults by IDSA-SHEA and ESCMID [100,102].

Patient Group	Recommendations		Alternative Treatment	
Initial episode	Fidaxomicin 200 mg	b.i.d., p.o., 10 days	Vancomycin 125 mg	q.i.d., p.o., 10 days
			Metronidazole 500 mg	t.i.d., p.o., 10–14 days
First recurrence	Fidaxomicin 200 mg	b.i.d., p.o., 10 days b.i.d., p.o., 5 days followed by q.o.d., p.o., 20 days	Vancomycin	p.o. tapered and pulsed regimen ^2^
			Vancomycin 125 mg	q.i.d., p.o., 10 days
			Bezlotoxumab ^3^ 10 mg/kg	I.V., once with SoC
Second or subsequent recurrence	Fidaxomicin 200 mg	b.i.d., p.o., 10 days b.i.d., p.o., 5 days followed by q.o.d., p.o., 20 days	Bezlotoxumab ^3^ 10 mg/kg	I.V., once, with SoC
	Vancomycin	p.o. tapered and pulsed regimen		
	Vancomycin 120 mg followed by rifaximin	q.i.d., p.o., 10 days followed by t.i.d., p.o., 20 days		
	fecal microbiota transplantation	Appropriate antibiotic treatment for at least 2 recurrences should be tried prior to offering fecal microbiota transplantation		
Fulminant ^1^	Vancomycin 500 mg	q.i.d., p.o., or by nasogastric tube (if ileus, rectal instillation)	-	-
	Metronidazole 500 mg	I.V., q.8h, together with p.o. or rectal vancomycin (if ileus)		

FMT, fecal microbiota transplantation; SoC, standard of care with antibiotics; p.o., oral administration; I.V., intravenous; q.d., once a day; b.i.d., twice a day; t.i.d., three times a day; q.i.d.; four times a day, q.8h, every eight hours; q.o.d., every other day. ^1^ Fulminant CDIs are defined by the presence of one of the following symptoms: hypotension, shock, ileus, and megacolon. ^2^ Tapered and pulsed vancomycin regimen includes 10- to 14-day q.i.d. oral administration of vancomycin 125 mg, 7-day b.i.d. oral administration of vancomycin, and then 2 to 8 weeks of q.2.d. to q.3.d. oral administration. ^3^ Using bezlotoxumab is considered an adjunctive treatment to standard of care, but the benefit of adding bezlotoxumab to fidaxomycin is unclear.

## Data Availability

Not applicable.

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
