# Peer review of "Gut Microbiota and New Microbiome-Targeted Drugs for Clostridioides difficile Infections"

_antibiotics, 2024, doi:10.3390/antibiotics13100995_

Round 1
Reviewer 1 Report
Comments and Suggestions for Authors
This is a very interesting review in which CDI -associated traits of gut microbiome and its metabolites were reviewed and microbiome -targeting treatment options are introduced, which was approved recently as a new drug by United States Food and Drug Administration (U.S. FDA), rather than a medical practice. However, the majority of the text refers to current and already known treatments. There is a lot of literature associated with this topic that has not been discussed in this review.
For example:
doi: 10.2217/fmb-2017-0203
Author Response
We acknowledge your comments. Together with the Gil et al., the list of references was up dated with more literatures.
Reviewer 2 Report
Comments and Suggestions for Authors
The manuscript reviewed some important aspects related to Clostridioides difficile infection and new-drugs based on gut microbiome specified by fecal microbiota transplantation. Authors concentrated to describe about the epidemiology, patholophysiology of C. difficile infections; gut microbiota and gut biotics related with CDI infection; CDI treatment therapies; introduced about fecal microbiota transplantation with advantages and disadvantages revealed from clinical trials. Many knowledges in this manuscript were well updated such as the trials of two approved fecal microbiota products. However, the manuscript remains some issues, authors please consider, revise and improve your manuscript;
1. The major problems
- Some parts, I think, authors must review more and described more detail, especially about the information related. As the indication, In the section "Epidemiology", authors should provide more about C. difficile incident rate, quantification of the disease occurrence by time (may be year); analysis of the disease distribution (populations, ages, when, where, ….); the morbidity rate, mortality rate, recirculation rate, …, the severity of the disease by time. The information helps reader understand about the significance and position of the disease.
- In the pathophysiology section, please provide detail in the regulatory mechanisms related to the C. difficile onset, development and outcome of disease. What is the bacteria evolution time by time, that related or unrelated to pathogenicity, toxicity of the bacteria; distribution of the high-pathogenic and multidrug resistance of the C. difficile strains. Based on the pathogenic mechanisms, what are targets for drug development.
- In the section 3. Please add more detail the structure of human microbiota at taxon of genera (in children, elders, and sensitive people) and emphasize to the distribution of Clostridium, C. difficile, threshold of C. difficile in healthy people, the shift of bacterial structures of human gut by ages and genders; bacteria inhibit C. difficile, the opportunity factors for C. difficile proliferation to be abundant; what are the C. difficile sensitive,… those all are basic for developing therapy such as fecal microbiome transplantation.
- The English in use and grammar in use in the manuscript must be improved by native English speaker, it had better to get assistance from a biologist.
2. Minor problems;
- Figure must cite references;
- Revise title of Table 2;
- Revise format of papers in the list of reference as the rule of the Journal;
- Many typo mistakes must be checked and fixed.

The English in use and grammar in use in the manuscript must be improved by native English speaker, it had better to get assistance from a biologist.
Author Response
- The major problems
- Some parts, I think, authors must review more and described more detail, especially about the information related. As the indication, In the section "Epidemiology", authors should provide more about C. difficile incident rate, quantification of the disease occurrence by time (may be year); analysis of the disease distribution (populations, ages, when, where, ….); the morbidity rate, mortality rate, recirculation rate, …, the severity of the disease by time. The information helps reader understand about the significance and position of the disease.
ANSWER> The epidemiology section was updated to have more information of the incidence rates, such as morbidity, with timelines, and disease distributions (lines 66-75), mortality rates (lines 62-65), recurrent rates (lines 78), and severity of the disease by time (line 109-114).
- In the pathophysiology section, please provide detail in the regulatory mechanisms related to the C. difficile onset, development and outcome of disease. What is the bacteria evolution time by time, that related or unrelated to pathogenicity, toxicity of the bacteria; distribution of the high-pathogenic and multidrug resistance of the C. difficile strains. Based on the pathogenic mechanisms, what are targets for drug development.
ANSWER> The pathophysiology section was updated to have the mechanisms of regulation of the C. difficile onset (lines 99-105), epidemiological clones with the characteristics (lines 80-91), toxicity of the bacteria with pathogenic mechanisms(lines 106-112), and characteristics of the clones (lines 80-91).
- In the section 3. Please add more detail the structure of human microbiota at taxon of genera (in children, elders, and sensitive people) and emphasize to the distribution of Clostridium, C. difficile, threshold of C. difficile in healthy people, the shift of bacterial structures of human gut by ages and genders; bacteria inhibit C. difficile, the opportunity factors for C. difficile proliferation to be abundant; what are the C. difficile sensitive,… those all are basic for developing therapy such as fecal microbiome transplantation.
ANSWER> The section 3 was updated to have details of the structure of human microbiota at taxon of genera and its association with age (line 140-145), C. difficile as a normal gut flora (line 146-149), and factors for C. difficile proliferations (line 198-207).
- The English in use and grammar in use in the manuscript must be improved by native English speaker, it had better to get assistance from a biologist.
ANSWER> English was edited with our best throughout the manuscript.
- Minor problems;
- Figure must cite references;
ANSWER> The figure was cited in line 247 and in line 255.
- Revise title of Table 2;
> Title of the table 2 was edited in line 354.
- Revise format of papers in the list of reference as the rule of the Journal;
ANSWER> List of the references was corrected.
- Many typo mistakes must be checked and fixed.
ANSWER> Typo was corrected throughout the manuscript.
Reviewer 3 Report
Comments and Suggestions for Authors
Comments to the manuscript “Gut microbiota and microbiome-targeted new drugs for Clostridioides difficile infections” by Lee et al.
The submitted manuscript reviewing the causes of gut microbiota dysbiosis and its relationships with Clostridioides difficile infection (CDI). The epidemiology, gut microbiota changes and metabolites, as well as treatment options for CDI are reviewed and explained.
The submitted review is well written, the subsections are properly described and includes the relevant information and bibliography for each issue of interest.
Thus, I consider that the manuscript is suitable for its publication in the Antibiotics Journal.
The only recommendation is including the figure number in line 198 and figure caption (line 225). Although it is the only figure in the manuscript, I suggest that it be numbered.
Author Response
The figure was made by our own analysis using data in clinicaltrials.gov. It was mentioned in the caption line 277. And the figure was numbered.
Round 2
Reviewer 1 Report
Comments and Suggestions for Authors
I agree with this new version
Author Response
Thank you so much.
Reviewer 2 Report
Comments and Suggestions for Authors
The manuscript was improved. The section 3.2 is incomplete, please check and finish.
Comments on the Quality of English LanguageEnglish should edited slightly.
Author Response
The manuscript was improved. The section 3.2 is incomplete, please check and finish.
> The incomplete words were removed.
English should edited slightly.
> Some sentences were edited. please find the track change.